# Finding a Landing Site in an Urban Area: A Multi-Resolution Probabilistic Approach

**DOI:** 10.3390/s22249807

**Published:** 2022-12-14

**Authors:** Barak Pinkovich, Boaz Matalon, Ehud Rivlin, Hector Rotstein

**Affiliations:** 1The Faculty of Computer Science, Technion Israel Institute of Technology, Haifa 3200003, Israel; 2Rafael Advanced Defense Systems Ltd., Haifa 3102102, Israel

**Keywords:** unmanned aerial vehicles, search theory, perception, semantic segmentation

## Abstract

This paper considers the problem of finding a landing spot for a drone in a dense urban environment. The conflicting requirements of fast exploration and high resolution are solved using a multi-resolution approach, by which visual information is collected by the drone at decreasing altitudes so that the spatial resolution of the acquired images increases monotonically. A probability distribution is used to capture the uncertainty of the decision process for each terrain patch. The distributions are updated as information from different altitudes is collected. When the confidence level for one of the patches becomes larger than a prespecified threshold, suitability for landing is declared. One of the main building blocks of the approach is a semantic segmentation algorithm that attaches probabilities to each pixel of a single view. The decision algorithm combines these probabilities with a priori data and previous measurements to obtain the best estimates. Feasibility is illustrated by presenting several examples generated by a realistic closed-loop simulator.

## 1. Introduction

Unlike conventional aircraft that takeoff and land on designated and controlled areas outside city limits, future commercial drones are expected to operate smoothly in crowded urban environments. Consequently, the well-defined zones delimited for landing, takeoff and safe flight will be replaced by dynamic and opportunistic areas within cities. To deal with this challenge, future delivery and transportation drones must be able to solve the “last-mile” problem (this refers to the last part in the delivery process of a product, namely, the section of transport from the last distribution center to the end customer), and find a place for landing with the following characteristics:Relatively close to the intended destination. These places cannot be limited to predetermined areas like heliports, sports fields, or similar places.Appropriate for the drone’s size and weight.Appropriate for landing under harsh (or relatively harsh) flight conditions compatible with the drone’s flying capabilities.Pose no safety concerns to itself, other vehicles, or living beings in the environment.

This paper describes a multi-resolution probabilistic approach to search for a landing place in a dense urban environment such as the one illustrated in Figure 1. In this context, multi-resolution is achieved by iteratively decreasing the altitude from which the visual sensor observes the urban environment, hence generating a sequence of images with monotonically increasing spatial resolution. Notice that the purpose is not to generate super-resolution images but rather to improve the level of confidence of the exploration. The result is probabilistic in the sense that confidence levels of different regions of the urban scenario are computed based on a priori knowledge and the result of observations. Somewhat related ideas for the different problem of performing an energy-efficient close inspection in an agricultural field were recently considered in [1].

## 2. Related Work

Although to the best of the author’s knowledge the problem considered in this paper is new, the probabilistic viewpoint provides connections with extensive existing literature, including search theory and Bayes-based decision-making. For example, Ref. [2] provides a survey on the usage of Bayesian networks usage for intelligent autonomous vehicle decision-making with no focus on specific missions. Similarly, Ref. [3] describes spacecraft autonomy challenges for future space missions, in which real-time autonomous decision-making and human-robotic cooperation must be considered. In a related autonomous spacecraft mission, Ref. [4] studies the selection of an appropriate landing site for an autonomous spacecraft on an exploration mission. The problem is formulated so that three main variables are defined on which to select the landing site: terrain safety, engineering factors (spacecraft’s descending trajectory, velocity, and available fuel), and the site preselected by using the available a priori information. The approach was tested using a dynamics and spacecraft simulator for entry, descent, and landing.

The problem considered here is also somewhat related to *forced landing*. This is because a UAV may need to decide the most suitable forced landing sites autonomously, usually from a list of known candidates [5]. In that work, references were made to the specifications for a forced landing system laid out in a NASA technical report essentially consisting of three main criteria: risk to the civilian population, reachability, and probability of a safe landing. The emphasis is on public safety, where human life and property are more important than the UAV airframe and payload. Specifications were included in a multi-criteria decision-making (MCDM) Bayesian network. See [6] for an application of the model to a real-life example. The initial design for UAVs’ autonomous decision systems for selecting emergency landing sites in a vehicle fault scenario are also considered in [6]. The overall design consists of two main components: preplanning and real-time optimization. In the preplanning component, the system uses offline information, such as geographical and population data, to generate landing loss maps over the operating environment. In the real-time component, onboard sensor data are used to update a probabilistic risk assessment for potential landing areas.

Another related field of interest is *search and rescue*, a challenging task, as it usually involves a large variety of scenarios that require a high level of autonomy and versatile decision-making capabilities. A formal framework casting the search problem as a decision between hypotheses using current knowledge was introduced in [7]. The search task was defined as follows: given a detector model (i.e., detection error probabilities for false alarms and missed detections) and the prior belief that the target exists in the search region, determine the evolution of the belief that the target is present in the search region as a function of the observations made until time *t*. The belief evolution is computed via a recursive Bayesian expression that provides a compact and efficient way to update the belief function at every time step as the searcher observes a sequence of unexplored and/or previously visited cells in the search region. After generating a method for computing the belief evolution for a sequence of imperfect observations, the authors investigated the search control policy/strategy. In the context of an area search problem, [1] investigated the uncertainty associated with a typical search problem to answer a crucial question: how many image frames would be required by a camera onboard a vehicle to classify a target as “detected” or “undetected” with uncertain prior information on the target’s existence. The paper presents a formulation incorporating uncertainty using beta distributions to create robust search actions. As shown below, these ideas are highly related to our approach.

Finally, ideas somewhat related to the ones considered here were used to pose a path-planning algorithm for performing an energy-efficient close inspection on selected areas in agricultural fields [8].

## 3. Problem Formulation

Suppose a drone needs to find a landing place in an urban area A. For simplicity, consider A to be planar, with an attached coordinate system {W} such that A lies on the x−y plane, measuring the altitude along the *z*-direction. Buildings and other constructions are modeled as occupied volumes over A. For example, the area considered in this paper will be 1 km by 1 km square. The drone can fly at different altitudes *h* while collecting measurements using a monocular camera and a visual sensor with range capabilities. Examples of the latter are an RGB-D sensor, a Lidar, or a couple of stereo cameras. Let $A_{h}$ be the plane parallel to A at an altitude *h* onto which A’s relevant characteristics can be mapped. For instance, a no-fly zone U in A (e.g., the base of a building) will be mapped onto the corresponding subset $U_{h}$ in $A_{h}$ (at least if *h* is smaller than the corresponding building’s altitude).

The mechanical structure of the drone, the size of its propellers, and the flying conditions (e.g., wind) constrain the minimum dimensions of the landing site on which the drone can safely land. Conservatively, A will be divided into a grid of identical cells $c_{ij}$, so that $\cup_{ij}c_{ij}=A$, and each $c_{ij}$ is in principle a landing site candidate. Note that this division is in-line with search theory (see, e.g., [9]) but stems from a different motivation: it is not a unit area being explored but the smallest area of interest. As discussed next, this will impact our development in several ways.

The images taken by a camera on the drone at time instant *t* will be a function of the pose $p_{t}$ and the camera’s field-of-view. Note that the camera’s orientation can differ from the drone’s by a relative rotation between the two. Assuming that the FOV is fixed and known, let $I_{t}\left(h\right)$ be the camera’s image at time *t* and $F_{t}\left(h\right)$ be the camera’s corresponding footprint, namely the 3D structure mapped onto the image plane. At the time *t*, the drone will have a unique pose $p_{t}$, but the dependence on the altitude is specifically considered in the notation; this is because the altitude scales the resolution and the footprint: for smaller *h*, one obtains a better resolution at the cost of a smaller footprint. The structure $F_{t}\left(h\right)$ is built on a collection of cells $C_{t}\left(h\right)={⋃}_{\left\{j,k\right\}\in \{J_{t}\left(h\right),K_{t}\left(h\right)\}}c_{ij}\subset A$.

Without additional constraints, the drone could fly over A at a relatively low altitude $h_{min}$, searching for an appropriate cell $c_{ij}$ on which to land. However, at this altitude, the camera’s *footprint*
$C_{t}\left(h_{min}\right)$ will include a relatively small number of cells $c_{ij}$ and hence the drone would spend a potentially prohibitive amount of time/energy exploring the whole A. On the other hand, the altitude can be selected to be the maximum allowable by regulations, say $h_{max}$, resulting in as large as possible footprints. In an extreme case, $C_{t}\left(h_{max}\right)=A$. This maximizes the area subtended by a single image and minimizes the exploration time, but will give rise to an image resolution that cannot guarantee the safety of landing, i.e., it will not resolve relatively small obstacles. The trade-off between altitude and resolution is solved in this work by considering a multi-resolution approach: the exploration will start at high altitude, say $h_{1}$, looking for the largest possible subset of A that appears to be *feasible* for landing, say $L_{1}$. Subsequently, the drone will reduce its altitude to $h_{2}$ and re-explore $L_{1}$ with the higher resolution resulting from $h_{2}<h_{1}$. This process results in a sequence of $A⊃L_{1}⊃\cdots ⊃L_{N}$ that will eventually converge to a collection of one or more safe landing places.

Figure 1, Figure 2 and Figure 3 illustrate this scenario.

### 3.1. A Probabilistic Model

In classical search theory [9], finding a target is often formulated as a decision problem by defining a set of binary variables: (1)$$H_{ij}≐\left\{\begin{array}{cc} 1 & \mathrm{if}\;c_{ij}\;\mathrm{has}\;a\;\mathrm{target}\\ 0 & \mathrm{otherwise} \end{array}\right.$$

This can be extended to the case of interest by defining: (2)$$H_{ij}≐\left\{\begin{array}{cc} 1 & \mathrm{if}\;c_{ij}\;\mathrm{is}\;\mathrm{appropriate}\;\mathrm{for}\;\mathrm{landing}\\ 0 & \mathrm{otherwise} \end{array}\right.$$

In the presence of uncertainty, each cell is likely to be suitable for landing with some probability $K_{ij}$ referred to as the *fitness for landing*. Clearly, if a cell is appropriate for landing, then $K_{ij}=1$, and if it is not, then $K_{ij}=0$. In real-world scenarios, prior knowledge about the fitness for landing can be based on using types of maps or 3D models that can be imprecise or outdated. Consequently, uncertainty in $K_{ij}$ needs to be incorporated into the model. Probably the simplest way to model the decision problem would be to introduce a binary distribution and say that the probability of $K_{ij}=1$ is *p* and $K_{ij}=0$ is 1−p. However, as observed by [1], this model fails to capture the *uncertainty* of the information, and instead, it is preferable to use a beta distribution for describing the prior knowledge together with its underlying uncertainty. Note that the beta distribution and the binomial and Bernoulli distributions form a *conjugate pair*, so that if a Bernoulli distribution can model the sensor, then the observation of new data changes only the parameters of the prior. At the same time, the conjugacy property ensures that the posterior is in the same class (i.e., beta). This is critical when propagating the belief in the fitness for landing in a Bayesian framework. Using these models simplifies the decision problem into a binary outcome as defined in Equation (Equation 2).

The beta distribution is defined as,
(3)$$Pr\left(K_{ij}\left|\alpha ,\beta \right.\right)=\frac{\Gamma (\alpha +\beta )}{\Gamma (\alpha )\Gamma (\beta )}K_{ij}^{\alpha -1}{\left(1-K_{ij}\right)}^{\beta -1}$$where $0<K_{ij}<1$ and Γ(α) is the gamma function defined as,
(4)$$\Gamma \left(\alpha \right)=\int_{0}^{\infty }x^{\alpha -1}e^{-x}\;dx$$

When α is an integer, Γ(α)=(α−1)!. The parameters α and β can be considered prior “successes” and “failures”. The beta distribution is somewhat similar to the binomial distribution. The main difference is that the random variable is $H_{ij}$ and the parameter is $K_{ij}$ in the binomial distribution, whereas the random variable is $K_{ij}$ and the parameters are α and β in the beta distribution.

Bayes theorem is often used in search theory [1,7,9] to update the aggregated belief (e.g., posterior distribution), which is proportional to the *likelihood* function times the prior distribution:(5)$$Pr\left(K_{ij}\left|S_{ij}^{N},\alpha ,\beta \right.\right)\propto Pr\left(S_{ij}^{N}\left|,K_{ij}\right.\right)Pr\left(K_{ij}\left|\alpha ,\beta \right.\right)$$Here, $S_{ij}^{N}=\sum_{n=1}^{N}H_{ij}^{n}$ is the number of successes in *N* Bernoulli trials. $Pr\left(K_{ij}\left|S_{ij}^{N},\alpha ,\beta \right.\right)$ is the posterior distribution for $K_{ij}$ given the number of successes. $Pr\left(S_{ij}^{N}\left|K_{ij}\right.\right)$ is the *likelihood function* and $Pr\left(K_{ij}\left|\alpha ,\beta \right.\right)$ is the prior distribution for $K_{ij}$. In [7], the likelihood distribution is given by a binomial distribution series of *N* observations:(6)PrSijNKij=SijNNKijSijN(1−Kij)N−SijN

Note that the underlying assumption is that the Bernoulli distribution provides an appropriate statistical model for the sensor used. This assumption is more or less natural when considering a series of noisy images. In our case, simple image processing algorithms are replaced by a more complex *meta*-sensor: fitness is computed by a semantic segmentation algorithm that associates for each pixel on a given image the suitability for landing on the corresponding cell on the ground. An appropriate model for this process is discussed next.

### 3.2. A Correlated Detection Model

The probability framework is motivated by the sensors’ limitations for establishing whether a given cell is appropriate for landing. As mentioned above, the main limitations are the camera resolution, the environmental conditions limiting visibility, and possibly scene dynamics. The probability estimated by a sensor that a given cell is appropriate for landing may vary according to altitude, with lower altitudes having higher confidence levels (up to a certain point depending on the sensor).

In [1], the authors assumed independent Bernoulli trials when detecting a target in recurrent visits. Independent trials are acceptable when the condition of the experiment does not change. However, in this research, our multi-resolution approach implies that when observing cell $c_{ij}$ at different altitude levels, one cannot assume uncorrelated measurements between different levels. At each level, the experiment’s condition changes (e.g., different resolution), and if a landing place exists, then it is expected that the rate of success will depend on previous trials and will increase when the level of details increases while descending toward cell $c_{ij}$.

Generalizing the binomial distribution typically involves modifying either the assumption of constant “success” probability and/or the assumption of independence between trials in the underlying Bernoulli process. The approach to generalizing the binomial distribution in this research follows the generalized Bernoulli distribution (GBD) model [10] by relaxing the assumption of independence between trials. The GBD model was further considered in statistics literature [11,12,13,14]) aiming to obtain its central limit theorems, including the strong law of large numbers and the law of the iterated logarithm for partial sums.

Consider a Bernoulli process $\left\{H_{ij}^{n},\;n\geq 1\right\}$ in which the random variables $H_{ij}^{n}$ are correlated so that the success probability for the trial conditional on all the previous trials depends on the total number of successes achieved to that point. More precisely, for some $0<K_{ij}<1$,
(7)$$Pr\left(H_{ij}^{n+1}\left|F_{ij}^{n}\right.\right)=\left(1-\theta_{ij}^{n}\right)K_{ij}+\theta_{ij}^{n}\frac{S_{ij}^{n}}{n}$$where $0\leq \theta_{ij}^{n}\leq 1$ are dependence parameters, $S_{ij}^{N}=\sum_{n=1}^{N}H_{ij}^{n}$ for N≥1 and $F_{ij}^{N}=\sigma \left(H_{ij}^{1},\cdots ,\;H_{ij}^{N}\right)$. If $H_{ij}^{1}$ has a Bernoulli distribution with parameter $K_{ij}$, it follows that $H_{ij}^{1},\;H_{ij}^{2},\cdots$ are identically distributed Bernoulli random variables.

By replacing the binomial distribution in Equation (Equation 5) with the GBD at each altitude $h_{n}$, the aggregated belief that a cell $c_{ij}$ is suitable for landing given the number of successes will be proportional to the product of the prior distribution and the altitude correlation-based distribution.

## 4. Testing Environment

To develop and test the probability multiple-resolution approach, a simulation environment was created using AirSim [15], a drone and car simulator built on the Unreal Engine [16]. AirSim is an open-source, cross-platform simulator for physically and visually realistic simulations. It is developed as an Unreal plugin that can be integrated into any Unreal environment. Within AirSim, a drone can be controlled using a Python/C++ API; for our project’s requirements, the drone can be configured similarly to a real drone in terms of dynamics, sensor data, and computer interface. The drone can be flown in the simulation environment from one waypoint to another while acquiring data from the sensors defined in the platform. The simulator computes images taken by a downwards-facing camera and a GPS/inertial navigation system for the current configuration.

The simulator has two primary purposes:To test the overall multi-resolution approach as the unit under test (UUT). The simulator functions as the hardware in the loop (HIL) tester’s data generator in this mode. The data generated by the simulator are streamed into the drone’s mission compute, and the system processes the data and computes the next coordinate to which the drone flies. Note that in this case, the simulator drives the real-time functioning of the closed-loop system.To generate offline data for the search algorithm. As mentioned above, the drone can be flown using the API around the map at different scenarios and heights while generating data at predetermined rates. Typical data consist of RGB images, segmented images, and inertial navigation data, forming a probabilistic model analysis and validation data set.

The 3D model used in the simulator was the Brushify Urban Buildings Pack [17], purchased from the Unreal marketplace. Figure 4 and Figure 5 show a simple example of RGB and corresponding segmentation images for the cameras simulated on the drone. The images highlight the observation that objects that occupy a cell are hardly detectable from a high altitude (e.g., a phone booth), while when descending, the gathered information allows the algorithm to detect these objects and decide that the drone cannot land in these specific cells.

## 5. Analysis and Preliminary Results

Obtaining prior knowledge about the urban scene is necessary for a probability model. For this purpose, a labeled 3D digital surface model (DSM) was generated by the simulation using a 2 × 2 [m] cell resolution.

The DSM, shown in Figure 6, allows for choosing the parameters of the prior distribution for each cell given the label of that cell. The labels chosen to be represented with initial probabilities were such that they were visible from a high altitude and may be considered appropriate for landing or not when descending. Figure 7 shows the beta distribution’s parameters for each label. These parameters were chosen to give some knowledge on an appropriate (or not) place to land. Still, there is sufficient uncertainty in the prior’s belief to allow some degree of freedom to change the values and the new belief with new observations.

The α and β parameters in Equation (Equation 3) are stored for each cell $c_{ij}$ in a 2D model of the urban scene, as illustrated in Figure 8. To determine if cell $c_{ij}$ contains a place to land, the probability $Pr(K_{ij}>\kappa )$ is calculated:(8)$$Pr\left(K_{ij}>\kappa \right)=\int_{\kappa }^{1}Pr\left(K_{ij}\right)\;dK$$

For instance, Figure 9 shows the prior belief that a landing place exists for $Pr(K_{ij}>0.5)$. Only sidewalks are somewhat appropriate for landing, even for κ=0.5, and the belief can easily be changed when new observations are obtained. Notice that cells in both Figure 8 and Figure 9 contain values for α, β, and prior probabilities.

The downwards-looking camera mounted on the drone provides color images that need to be converted into information on how suitable each cell is for landing. Clearly, this relation may be highly complex. In recent years, deep neural networks have shown to be successful in various computer-vision applications, including the type of semantic segmentation problems relevant to our purpose. Consequently, we chose to employ the semantic segmentation network BiSeNet [18], which fuses two information paths: context path and spatial path. The context allows information from distant pixels to affect a pixel’s classification at the cost of reduced spatial resolution. In contrast, the spatial path maintains fine details by limiting the number of down-sampling operations. This net also provides a reasonable compromise between segmentation accuracy and compute requirements.

Other network architectures, potentially more complex and accurate, are currently under investigation, including DeepLabV3+ [19] (a state-of-the-art convolutional net that combines an atrous spatial pyramid with an encoder–decoder structure) and Segformer [20] (a relatively efficient transformer-based segmentation model). Preliminary results when using these semantic segmentation architectures on real-life data are discussed in a forthcoming paper currently under review.

BiSeNet was trained and validated on images taken by the camera while flying in the urban environment at different altitudes. The model uses the labels in the digital surface model in Figure 6 for training. During inference, the model predicts probability scores (summing to 1) for the different categories using captured images. Each category is also assigned an a priori weight representing how suitable this category is for landing (e.g., weight[sidewalk] = 0.8 and weight[building] = 0). We take a weighted average of the categories’ probabilities using the predefined weights to obtain a final score $p_{mn}$ for each image pixel. $p_{mn}$, which can vary between 0 to 1, which describes the probability that a landing site exists based on the observed data. Using the 6-DOF of the drone, the image footprint, i.e., pixels coordinates projected on the ground, was transformed to a world coordinate system to be associated with each cell $c_{ij}$ in the grid. The outcome of a Bernoulli trial for success or failure is given by counting $N_{p}$, the number of pixels associated with cell $c_{ij}$ that pass $p_{mn}>0.5$ and are relative to $N_{pc}$, the total number of pixels associated with $c_{ij}$. If the relative amount is greater than 0.99, then $c_{ij}$ holds a successful trial.
(9)$$N_{p}=\sum_{m,n\in i,j}^{N_{pc}}\left\{p_{mn}>0.5\right\}$$
(10)$$H_{ij}≐\left\{\begin{array}{cc} 1, & \frac{N_{p}}{N_{pc}}>0.99\\ 0, & \mathrm{otherwise} \end{array}\right.$$

### 5.1. A Single-Altitude Bayesian Update

The Bayesian update was tested for several flight scenarios. Suppose now that the drone flies and takes images at a constant altitude and that the semantic segmentation network analyzes images. Each cell $c_{ij}$ belonging to an image footprint is associated with the corresponding projected pixels, and a Bernoulli trial is performed according to Equation (Equation 10). The trial is performed on each cell only once to prevent added correlation effects at that altitude. The outcome of the single trial is added to the α and β values previously selected as the prior (shown in Figure 7 and Figure 8) and then integrated numerically according to Equation (Equation 8). Figure 10 and Figure 11 show the updating stage at different time instances and altitudes.

### 5.2. An Altitude-Based Bernoulli Trial Distribution

In order to study the GBD model, an experiment with Bernoulli trials at different altitudes was performed. Several objects are placed on the ground at different locations around the urban scene. The experiment was planned so that a single object was selected for the drone to descend upon at different locations. At each location, images are taken as input for the semantic segmentation network. On each output of the network, a Bernoulli trial is performed on a single cell according to Equation (Equation 10), so that at each altitude that an image is taken, there is a single success or failure output on a given cell $c_{ij}$. Figure 12 and Figure 13 show the input and output at selected altitudes in a single location. A phone booth was selected for the drone to descend upon. There are 31 locations around the urban scene with the phone booth placed on the sidewalk. A cell in the world coordinate system was selected at each location so that the phone booth occupies partially or the entire cell. The expected outcome is that the phone booth would be partially detected at high altitudes, and the selected cell would be detected as fit for landing. In contrast, more details will be detected when descending, and the cell will be detected as unfit for landing.

Figure 14 shows the histogram for the 31 locations with altitude-based trials. We can see that the aforementioned expected behavior is indeed observed under approximately 160 m. Peculiarly, above this altitude, the success frequency diminishes markedly. This may be explained by the fact that there were no images from these altitudes in the training set, making the net’s prediction unreliable. It should be noted that even in lower altitudes, its prediction can be noisy for various reasons, mainly because the training set is not diverse enough. We expect that more diversity in the training set will yield greater reliability of the net, which in turn, will result in a better fit for the model.

## 6. Conclusions and Further Work

This paper has presented a multi-resolution probabilistic approach to finding an appropriate landing site for drones in a dense urban environment. The approach uses a priori data (e.g., a map or a DSM) to estimate the fitness for landing probability distribution for each cell on which the environment is divided. Distribution and not probabilities are used in an attempt to model the uncertainty of the data. Subsequently, the data collected by a visual sensor and processed by a semantic segmentation neural net are used to update the distribution using Bayesian networks. In order to do that, the probability of success is factored into the results obtained by the net. Images are captured at different altitudes in an attempt to solve the trade-off between image quality, including spatial resolution, context, and others. After presenting theoretical aspects, the simulation environment in which the approach was developed and tested is detailed, and the experiments conducted for validation are described. The overall approach is shown to produce the desired results, at least for the simulation environment in which it was tested.

Further work is currently underway in three main directions. Firstly, we would like to establish some success criteria for the procedure. For instance, we would like to develop bounds to enable more accurate ways of analyzing our results. Secondly, we would like to generate more realistic images on which semantic segmentation can be trained and tested. Lastly, we would like to test the approach on actual data and conduct a flight test to achieve real-life validation.

## Figures and Tables

**Figure 1 sensors-22-09807-f001:**
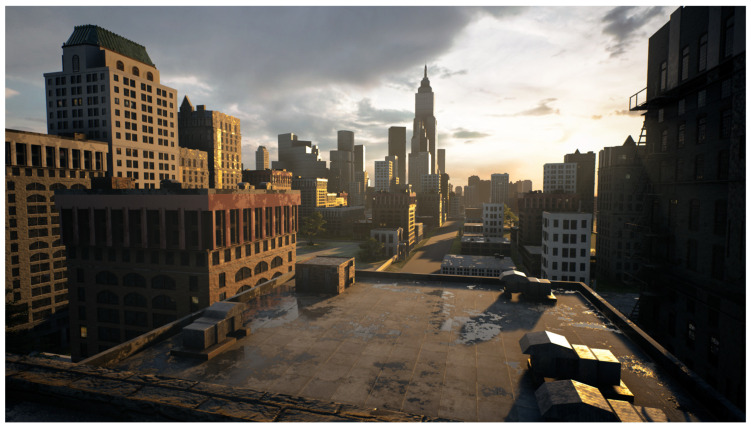
Simulated urban environment in which the drone attempts to land.

**Figure 2 sensors-22-09807-f002:**
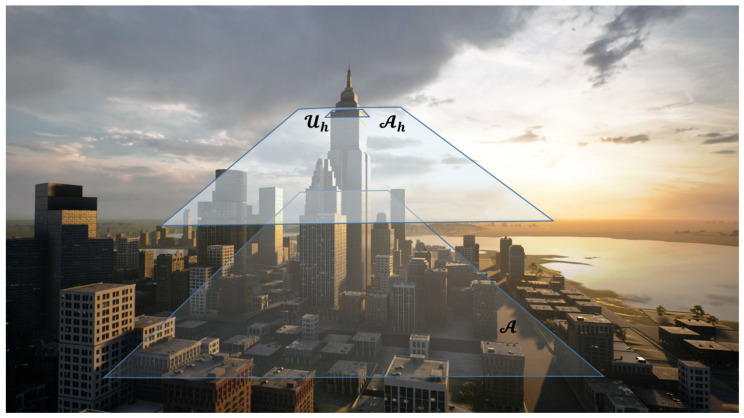
The area of interest A, the plane $A_{h}$ at an altitude *h*, and the mapping of an obstacle $U_{h}$.

**Figure 3 sensors-22-09807-f003:**
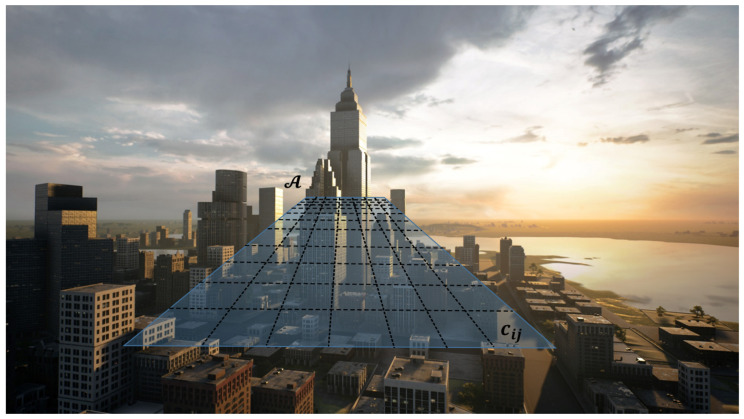
The plane at a given altitude is divided into small cells.

**Figure 4 sensors-22-09807-f004:**
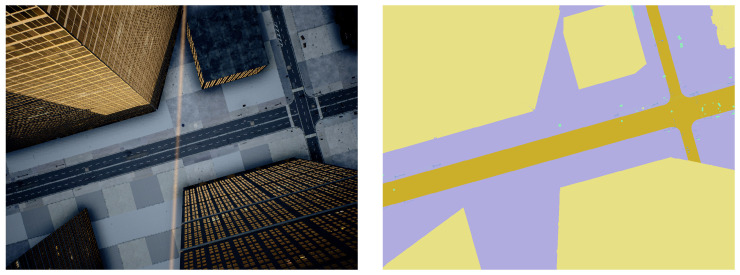
High-altitude urban scene captured with a downward-looking camera. On the left is a simulated image. On the right is the corresponding segmented scene.

**Figure 5 sensors-22-09807-f005:**
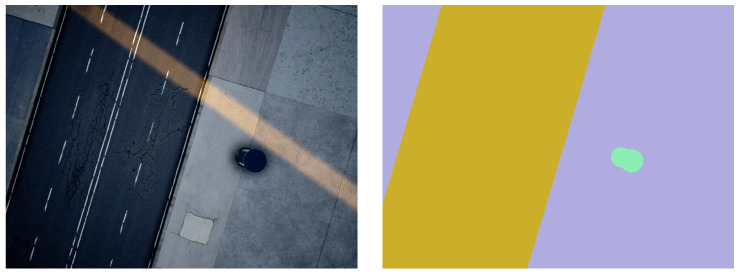
Low-altitude urban scene captured with a downward-looking camera. On the left is a simulated image. On the right is the corresponding segmented scene.

**Figure 6 sensors-22-09807-f006:**
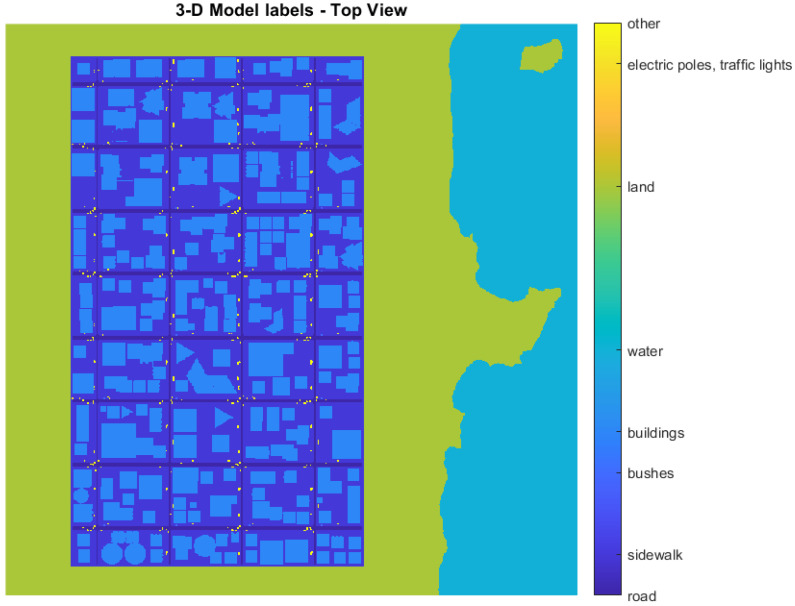
Top view of the digital surface model with chosen labels.

**Figure 7 sensors-22-09807-f007:**
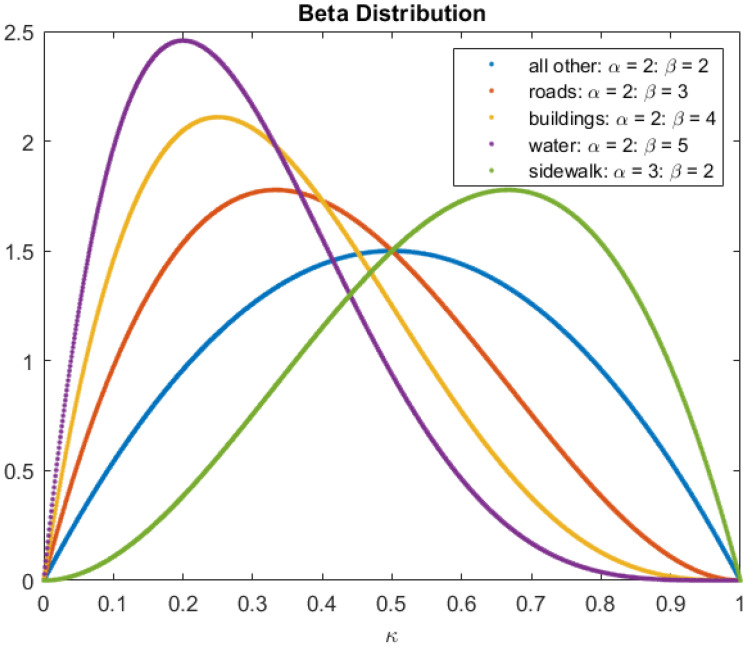
Prior beta distribution for chosen labels.

**Figure 8 sensors-22-09807-f008:**
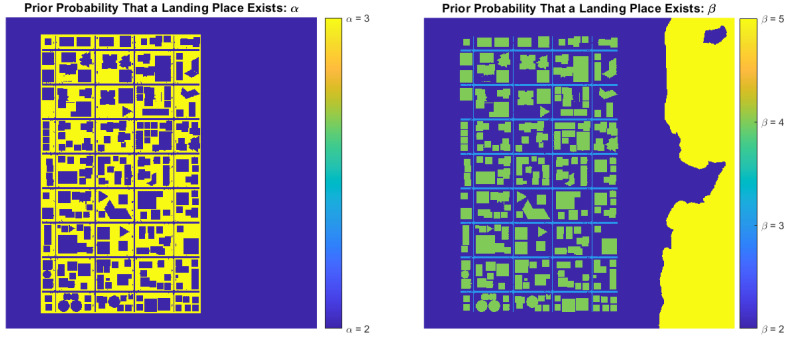
Prior α and β parameters for each cell in the urban scene.

**Figure 9 sensors-22-09807-f009:**
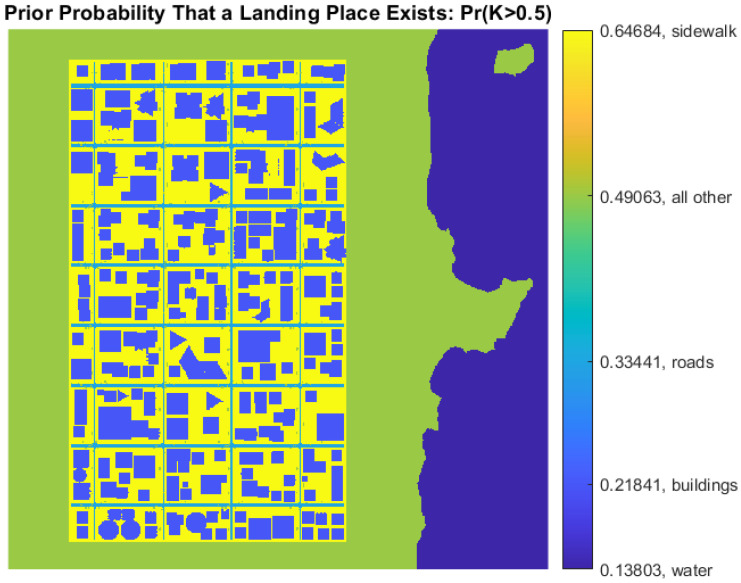
Prior landing probability for κ=0.5.

**Figure 10 sensors-22-09807-f010:**
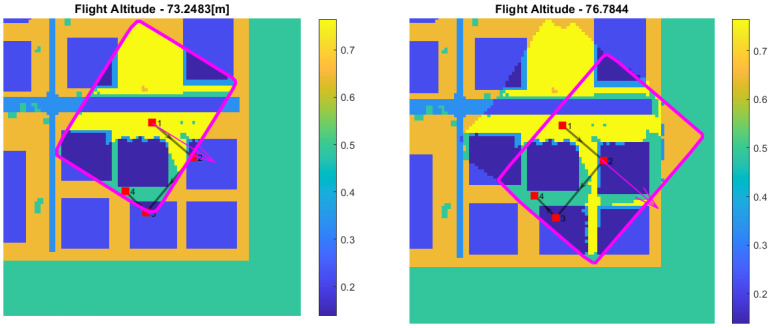
Bernoulli trial update: each cell is updated only once.

**Figure 11 sensors-22-09807-f011:**
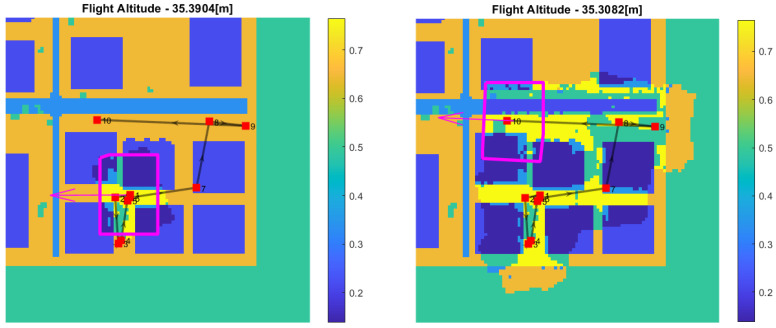
Bernoulli trial update: each cell is updated only once.

**Figure 12 sensors-22-09807-f012:**
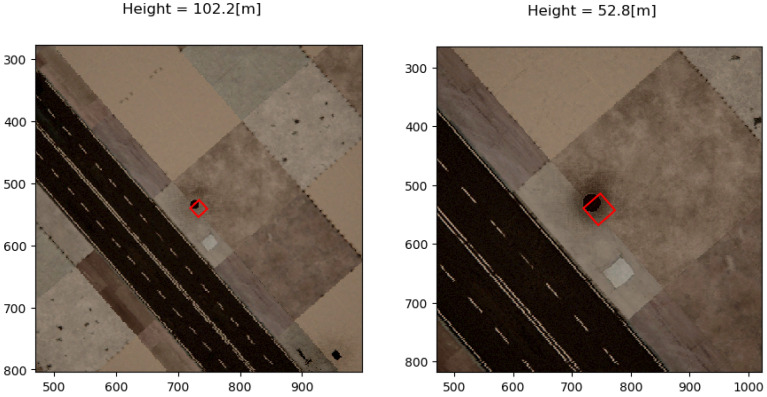
Semantic segmentation input: telephone booth partially occupies the cell.

**Figure 13 sensors-22-09807-f013:**
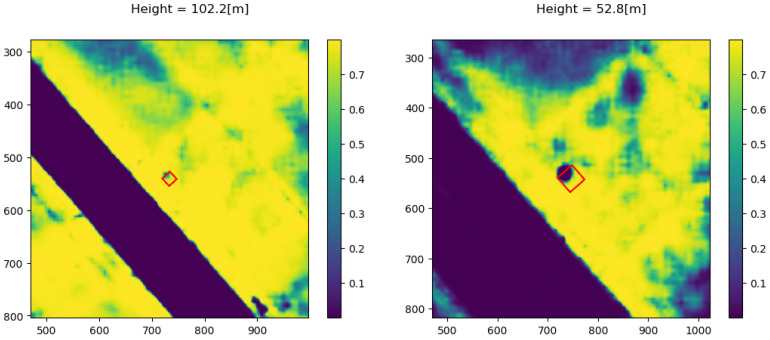
Semantic segmentation output: telephone booth partially occupies the cell.

**Figure 14 sensors-22-09807-f014:**
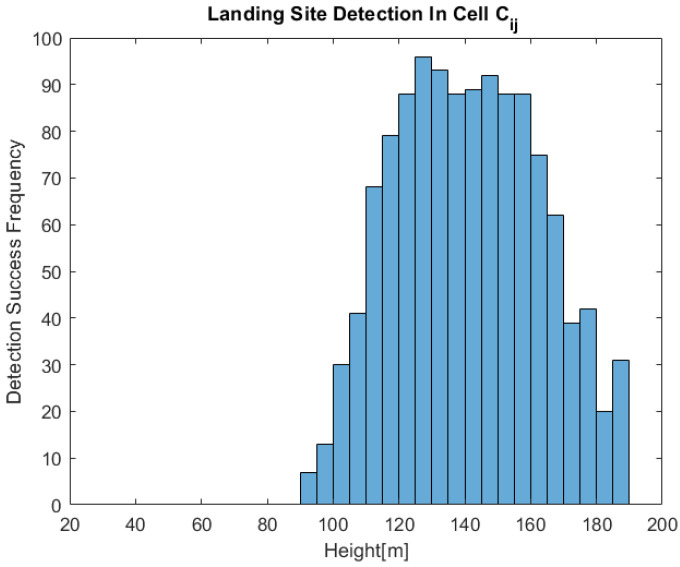
An altitude-based Bernoulli trial histogram.

## Data Availability

Not applicable.

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
