# Peer review of "Finding a Landing Site in an Urban Area: A Multi-Resolution Probabilistic Approach"

_sensors, 2022, doi:10.3390/s22249807_

Round 1
Reviewer 1 Report
The topic is quite interesting. However, the following issues should be addressed before the full acceptance: 1. line 56: the citation is malformed. 2. line 231: the semantic segmentation network BiSeNet is relatively old (2018). The authors should discuss and compare some new segmentation networks that have been published recently. 3. The authors should add the original images obtained by the camera from the top view for Fig. 6, 8, and 9. 4. The authors should add comparisons with other landing site selection methods (if any).Author Response
Thank you very much for your constructive reviews. We have prepared an updated version taking into account your comments.
1. The citation was corrected. 2. The purpose of the paper was to present an overall approach to solving the problem and not to discuss performance using different semantic segmentation architectures. This problem is indeed very interesting and was pursued in additional work on real images which is currently under review for another publication. It is worth stressing that we believe that the overall approach and specific semantic segmentation selections are independent issues. 4. We have stressed that figures 4, 8 and 9 contain 3D models with semantic and statistic information, not coming from camera images. At this point, we preferred not to add additional images (although this indeed easy to get from the simulation) since they would enlarge the paper without providing additional clarity. 5. To our knowledge there is no further method for solving our problem.
Reviewer 2 Report
The authors present a multi-resolution probabilistic approach to finding a landing site in an urban area. The method is tested using a simulation environment created using AirSim and utilized the Unreal Engine AirSim.
The document requires much work, from clearly defining the problem to a better paper presentation. For instance, it needs to be clarified what the authors mean by a multi-resolution approach; even in one part, they talk about an RGB-simulated camera which should be considered color images. Next, the problem definition requires improvement.
For the first part, please consider the following:
· The title is too general. Is an approach using what?
· The italics in the abstract are unnecessary.
· Again, in the abstract only ‘a multi-resolution approach’ is indicated with no further details. What frequencies? What bandwidths are considered?
· The abstract lacks quantitative details and the most relevant result.
· Do you have any reference for the term “the last-mile problem”? Is this term defined by these authors only?
· If [1] presents “similar ideas,” as stated by the authors, why do we need another approach?
· What is the goal of Figure 1?
· Is [2] generally used as background and state-of-the-art comparison?
· Figure 2 is difficult to understand for the indicated purposes—a similar problem to Figure 3. I recommend changing Figures 1-3 to others that could better help explain their ideas. Even a simulated photo like the one used could work if simulated within the day with daylight. The current images are dark and combined with black color text. Also, recall that the figures need to explain something; it’s not only about writing titles.
Author Response
Please see the file attached

Round 2
Reviewer 1 Report
Since most of my comments are properly covered, I think the revision can be accepted in its current form.